# Whole-Genome Sequencing of Six Neglected Arboviruses Circulating in Africa Using Sequence-Independent Single Primer Amplification (SISPA) and MinION Nanopore Technologies

**DOI:** 10.3390/pathogens11121502

**Published:** 2022-12-08

**Authors:** Ansgar Schulz, Balal Sadeghi, Franziska Stoek, Jacqueline King, Kerstin Fischer, Anne Pohlmann, Martin Eiden, Martin H. Groschup

**Affiliations:** 1Institute of Novel and Emerging Infectious Diseases, Friedrich-Loeffler-Institut, 17493 Greifswald-Insel Riems, Germany; 2Institute of Diagnostic Virology, Friedrich-Loeffler-Institut, 17493 Greifswald-Insel Riems, Germany

**Keywords:** MinION sequencing, SISPA, arboviruses, Africa

## Abstract

On the African continent, a large number of arthropod-borne viruses (arboviruses) with zoonotic potential have been described, and yet little is known of most of these pathogens, including their actual distribution or genetic diversity. In this study, we evaluated as a proof-of-concept the effectiveness of the nonspecific sequencing technique sequence-independent single primer amplification (SISPA) on third-generation sequencing techniques (MinION sequencing, Oxford Nanopore Technologies, Oxford, UK) by comparing the sequencing results from six different samples of arboviruses known to be circulating in Africa (Crimean–Congo hemorrhagic fever virus (CCHFV), Rift Valley fever virus (RVFV), Dugbe virus (DUGV), Nairobi sheep disease virus (NSDV), Middleburg virus (MIDV) and Wesselsbron virus (WSLV)). All sequenced samples were derived either from previous field studies or animal infection trials. Using this approach, we were able to generate complete genomes for all six viruses without the need for virus-specific whole-genome PCRs. Higher Cq values in diagnostic RT-qPCRs and the origin of the samples (from cell culture or animal origin) along with their quality were found to be factors affecting the success of the sequencing run. The results of this study may stimulate the use of metagenomic sequencing approaches, contributing to a better understanding of the genetic diversity of neglected arboviruses.

## 1. Introduction

Recent pandemics have illustrated that emerging and re-emerging infectious diseases are of utmost importance for the global population. Despite not being a novel phenomenon, the worldwide transport of passengers and cargo, extensive land use, and the ongoing increase in the world population combined with urbanization and deforestation are favoring the emergence and accelerating the spread of pathogens [1]. Most of the emerging infectious diseases are caused by zoonotic pathogens, with the importance of vector-borne diseases having increased greatly in recent decades [2]. Especially in Africa, there are a large number of (neglected) arthropod-borne viruses (arboviruses) with zoonotic potential, and yet little is known of most of these viruses regarding their actual distribution, life cycle, host ranges, and genetic diversity [3]. Therefore, it is of major importance to investigate these tropical arboviruses in order to reduce the threat they pose to human and animal health and to proactively prevent large-scale emergence [2]. Alongside reliable molecular diagnostics such as RT-qPCRs, producing longer sequence reads (up to the full genomes) is essential for the phylogenetic characterization of viruses. First-generation sequencing (e.g., Sanger sequencing) usually only allows the sequencing of shorter amplicons of the target pathogen. In this context, and also for the discovery of new pathogens, next-generation sequencing (NGS) techniques are attracting increasing interest [4].

NGS plays an essential role in identifying novel genomes and analyzing epigenetic factors. Its innovation, effectively utilized and optimized over the past decade, has revolutionized the genomic investigation of humans as well as animals, plants and microorganisms [5,6]. Most of the available second-generation sequencing methods (ion semiconductor, pyrosequencing and sequencing by synthesis) generate short reads (30–800 bp). Consequently, there are some limitations with these technologies, such as in the assembly and determination of complex genomic regions and in the detection of DNA methylation and gene isoforms [7,8]. In recent years, third-generation sequencing (TGS) has been developed to overcome these challenges. It enables the generation of long reads and has demonstrated its competence in whole-genome sequencing for several pathogens [9]. In this regard, one of the most effective and efficient TGS instruments for real-time identification of a broad range of viruses is the MinION sequencing device (Oxford Nanopore Technologies, Oxford, UK). In contrast to other NGS instruments, the MinION platform is comparatively cost-effective, and because of its small size and portability, this technology is also suitable for research in the field [10].

However, one of the crucial steps usually required for NGS is the enrichment of the viral genomes in samples prior to sequencing. In this context, “sequence-independent single primer amplification” (SISPA), a method based on nonspecific amplification using random primers [11], represents a universally applicable and already proven approach for NGS. SISPA was first developed by Reyes and Kim (1991). The first step of it is a reverse transcription, where random hexamers labeled with a known specific sequence are directly incorporated into the cDNA. After denaturation, annealing and double-strain synthesis (Klenow reaction), the yielded dsDNA is amplified using the second SISPA primer consisting of the corresponding known specific sequence without the random hexamers. Hence, it allows the enrichment of the viral genome without the need for virus-specific primers [12,13,14]. Although the output generated by SISPA is generally lower compared to a virus-specific whole-genome PCR, the variable application range poses a major advantage. Thus, this enrichment method is particularly suitable for sequencing viruses originating from different genera, requiring relatively little effort. So far, SISPA in combination with MinION nanopore sequencing (TGS) has only been performed for a limited number of viruses, e.g., canine distemper virus [15], enteroviruses [16], African horse sickness virus [17], Jingmen tick virus and Crimean–Congo hemorrhagic fever orthonairovirus [18]. 

The main objective of the present study was to compare the number of specific sequencing reads of six selected arboviruses circulating in Africa using SISPA-based nanopore sequencing under different preconditions. Aside from Crimean–Congo hemorrhagic fever virus (CCHFV) and Rift Valley fever virus (RVFV), representing two very common and severe agents of zoonotic diseases in Africa, less well-studied pathogens such as Nairobi sheep disease virus (NSDV), Middelburg virus (MIDV), Dugbe virus (DUGV) and Wesselsbron virus (WSLV) were also sequenced as part of the study. Especially for the four last-mentioned viruses, scientific data are extremely limited [3], and there are only a few (whole-) genome sequences available in public databases (INDSC). RVFV (genus *Phlebovirus*), CCHFV, DUGV and NSDV (genus *Orthonairovirus*) belong to the order *Bunyavirales* and thus are RNA viruses characterized by a single-stranded tripartite genome. Their three genome segments are divided into a small (S-) segment of 1–2 kb, a medium (M-) segment of 3.7–5 kb, and a large (L-) segment varying from 6.8 to 12 kb [19]. In comparison, MIDV (genus *Alphavirus*) and WSLV (genus *Flavivirus*) have an unsegmented, single-stranded RNA genome comprising approximately 11 kb [20,21]. In order to have a practical and easy-to-use benchmark for sample quality, which can also be applied in laboratories with only basic infrastructure, the sequencing results of samples of the same type but with different quantification cycle (Cq) values in diagnostic RT-qPCR were compared for two selected viruses. 

The results of this study may contribute to, as well as encourage, the generation and provision of viral sequences of neglected tropical arboviruses, allowing a better understanding of their genetic diversity and distribution.

## 2. Materials and Methods

### 2.1. Virus Samples, Metadata and Cultivation

Six different viruses belonging to four different genera were analyzed by nanopore sequencing. For four of them, two different types of samples were tested and compared: samples of animal origin (vectors or hosts) from field studies or experimental animal trials and samples derived from cell culture. Moreover, for two selected viruses (CCHFV and RVFV), the obtained number of specific reads was compared for three different Cq values of the samples. 

RNA extraction of in vitro samples was performed using the QIAamp Viral RNA Mini Kit (Qiagen, Hilden, Germany). The extraction of RNA from samples of animal origin was conducted using the NucleoMag^®^ VET kit (MACHEREY-NAGEL GmbH &Co. KG, Düren, Germany) and a King Fisher extraction device (Thermo Fisher Scientific, Waltham, MA, USA) according to the manufacturer’s instructions. 

#### 2.1.1. CCHFV

The African strain Ibar10200 (Africa-I lineage) was grown on Vero E6 cells (African green monkey kidney cells, Collection of Cell Lines in Veterinary Medicine, Friedrich-Loeffler-Institut, FLI; CCLV-RIE 0929) under biosafety level (BSL)-4 conditions. Analysis of the extracted sample by RT-qPCR [22] revealed a Cq value of 21. The CCHFV field samples originated from a study conducted in Mauritania in 2018 [23]. In this survey, *Hyalomma* ticks were collected from cattle and camels. The ticks were individually homogenized in AVL lysis buffer, and then RNA extraction was performed. For a better assessment of the influence of the Cq value on the quality of the sequencing output, three positive ticks with different Cq values (19; 26; 30) were selected for sequencing. The three field samples belonged to the lineages Africa I and III.

#### 2.1.2. RVFV

The live-attenuated RVFV vaccine strain MP-12 was grown on Vero 76 cells (African green monkey kidney cells, Collection of Cell Lines in Veterinary Medicine, FLI; CCLV-RIE 0228). After heat inactivation in AVL lysis buffer at 70 °C for 10 min, RNA extraction of the supernatant was performed. The following RT-qPCR [24] yielded a Cq value of 20. Furthermore, RNA from positive tissue samples originating from black rats (*Rattus rattus*) that were infected with RVFV strain 35/74 under BSL-3 laboratory conditions [25] was used as samples of animal origin. As for CCHFV, three samples with three different Cq values (lungs 20, kidney 24, and spleen 30) were applied for nanopore sequencing. 

#### 2.1.3. DUGV

The Nigerian DUGV prototype strain IbAr 1792 (kindly provided by the World Reference Center for Emerging Viruses and Arboviruses, University of Texas Medical Branch (UTMB), Galveston, TX, USA) was grown on SW13 cells (human adrenal gland cells, kindly provided by Karolinska Institutet, Solna, Sweden), and RNA was extracted from the virus-containing supernatant. A Cq value of 15 was determined by RT-qPCR [26]. The field sample originated from a DUGV-positive *Amblyomma* tick collected in 2018 in Nigeria [27]. The tick was individually homogenized in AVL lysis buffer before RNA extraction was conducted, and RT-qPCR showed a Cq value of 20. 

#### 2.1.4. NSDV

The NSDV strain IG619 (kindly provided by the World Reference Center for Emerging Viruses and Arboviruses, UTMB, Galveston, TX, USA) was grown on SW13 cells. In RT-qPCR [28], the extracted RNA of the cell culture supernatant showed a Cq value of 15. A positive tissue (bovine liver) sample from an NSDV animal infection trial conducted under BSL-3 laboratory conditions [28] was used as a sample of animal origin. This sample had a Cq value of 23. 

#### 2.1.5. MIDV and WSLV

The MIDV strain MT MP160 and the WSLV strain SA H177 (kindly provided by the World Reference Center for Emerging Viruses and Arboviruses, UTMB, Galveston, TX, USA) were grown on BHK-21 cells (baby hamster kidney cells, Collection of Cell Lines in Veterinary Medicine, FLI, CCLV-RIE 0164), and RNA extraction was performed from the virus-containing supernatants. By using RT-qPCR (Appendix A), Cq values of 19 (MIDV) and 25 (WSLV) were obtained. Since no positive field samples or material from experimental infection trials of those viruses were available, only the cell culture supernatants could be applied for sequencing.

### 2.2. SISPA and Sample Preparation for Nanopore Sequencing

The SISPA methodology was carried out using primers and PCR conditions as outlined in the protocol published by Peserico et al. [15]. In the RT step, the first SISPA primer (GCCGGAGCTCTGCAGATATCNNNNNN) and nNTPs (1 μL each) were mixed with 11 μL of viral RNA and incubated at 65 °C for 5 min. Afterwards, a second master mix (4 μL SSIV buffer 5×; 1 μL DTT; 1 μL RNase Inhibitor; and 1 μL SSIV Reverse Transcriptase (SuperScript IV Reverse Transcriptase Kit; Invitrogen, Waltham, MA, USA)) was added and incubated (23 °C for 10 min; 50 °C for 50 min; 80 °C for 10 min/one cycle each) in a GeneTouch Plus Thermal Cycler (Biozym Scientific GmbH, Hessisch Oldendorf, Germany). Double-strain synthesis was performed by adding 1 μL of a Klenow polymerase (New England Biolabs, Ipswich, MA, USA) under the following conditions: 37 °C for 60 min and 75 °C for 10 min. The amplification of 5 μL of ds cDNA was carried out after adding the third master mix (5 μL 10× PfU Ultra II reaction buffer; 1 μL PfU Ultra II Fusion HS DNA Polymerase (both Agilent, Santa Clara, CA, USA); 1.25 μL dNTPs (Invitrogen, Waltham, MA, USA); 1 μL of the second SISPA primer; and 36.75 μL nuclease-free water). Hereby, the following temperature profile was used: initial denaturation for 1 min/95 °C; DNA denaturation for 20 s/95 °C; annealing for 20 s/65 °C; extension for 3 min/72 °C; and final extension for 3 min/72 °C. DNA denaturation, annealing and extension were repeated for 45 cycles. Moreover, an additional SISPA primer (GACCATCTAGCGACCTCCACNNNNNNNN) by Chrzastek et al. [29] was used in the same concentrations and quantities as described in the protocol above [15]. The difference between these primers lies in the length of the 5′ tag N (6 N vs. 8 N) of the binding site; while the barcode length is identical for both (20 bp), they differ in their sequence. In order to evaluate whether one primer set is more suitable for the generation of specific reads, each virus sample was sequenced individually using one of the two SISPA primer sets. After the SISPA amplification step, all samples were purified using AMPure XP magnetic beads (Beckman Coulter, Brea, CA, USA) in an ×1.8 sample volume to bead volume ratio, followed by a sample library preparation for MinION sequencing according to a previously published and adapted protocol [30]. This protocol combines the SQK-LSK109 kit with the EXP-NBD104 kit (both from Oxford Nanopore Technologies, Oxford, UK) to allow simultaneous sequencing of multiple samples. The prepared library was spotted onto a Flow Cell R9.4.1 (FLO-MIN106D, Oxford Nanopore Technologies, Oxford, UK) and sequenced with a MinION Mk1C instrument (Oxford Nanopore Technologies, Oxford, UK). Figure 1 provides an overview of the workflow. Sequencing was run for at least 48 h until all pores of the flow cell were depleted. For each run, six to ten different barcoded samples were sequenced. Usually, only reads of barcodes dedicated to a specific sample are used to build up the consensus sequence, whereas the unclassified reads (all reads that were unable to be assigned to any of the barcodes used) are not included in the evaluation. Those unlabeled reads represent a potpourri of DNA sequences of all samples in one sequencing run and thus can be used to search for additional virus-specific reads. Since two samples of the same virus from different origins were never included within the same run, both the classified and unclassified reads could be evaluated.

### 2.3. Analysis of MinION Sequence Data

The steps of the data analysis are shown in Panel VI “Evaluation” of Figure 1. Base-calling is the initial process of assigning nucleobases to electrical current changes as a result of nucleotides passing through the nanopores. Raw signals (Fast5 raw data reads) are translated into nucleotide sequences, and these sequences are provided for downstream analysis. After that, reads are demultiplexed (NGS reads are assigned to the sample of their corresponding barcode) and trimmed (removal of adapter sequences and low-quality bases). In this study, Fast5 raw data reads produced by the arbovirus libraries were base-called (high accuracy), demultiplexed and trimmed using the Mk1C sequencer (Guppy version 3.2.10, Oxford Nanopore Technologies). Additional demultiplexing and adaptor removal were performed using Porechop on the NanoGalaxy platform [31]. Base-called and demultiplexed sequencing data quality was assessed with NanoPack (version 1.13.0, https://github.com/wdecoster/NanoPlot; accessed on 1 August 2022). Reads with a minimum quality of 7 were considered for further analysis. For consensus sequence generation from trimmed FastQ reads, alignment against redundant databases and mapping with reference genomes (version 20, https://rvdb.dbi.udel.edu/; accessed on 3 September 2020) were performed using k-mer alignment (KMA) [32] and Minimap2 [33]. The KMA readouts were used for computing the genome coverage and accuracy of the consensus sequence.

## 3. Results

Table 1 provides an overview of the total and the virus-specific read counts that were received for all six arboviruses (CCHFV, RVFV, NSDV, DUGV, WSLV and MIDV), including outcomes for the different origins (animals and cell culture). The read counts obtained by using the two different SISPA primer sets [15,29] and the total number of reads (including both primer sets as well as unclassified reads) are given for each sample.

Important quality parameters of the sequencing results (coverage, depth, read quality, read length and identity levels) of all samples are summarized in Table 2. Likewise, Table 2 includes the results obtained with the two different SISPA primer sets for preamplification, as well as the complete results, i.e., the results obtained with the two primer sets combined with the unclassified reads. 

The generated genome sequences are deposited in Appendix A. The highest numbers of reads were found for unsegmented viruses, namely MIDV (400,767) and WSLV (296,123; both Table 1). On the other hand, considerably lower read numbers were obtained for the examined segmented bunyaviruses (DUGV L-segment: <57,423; CCHFV L-segment: <6691; NSDV L-segment: <4696; RVFV L-segment: <2567) (Table 1). By using the described protocol [15], the primers of Chrzastek et al. [29] showed less efficiency compared to the primer set of Peserico et al. [15]. The best results were achieved by combining the sequencing results of the two different primer sets for the respective samples (Table 1 and Table 2). Using the unclassified reads, the output of specific reads could even be enhanced to a total of 13–57% (Table 1, P+C+U). The data obtained for the different CCHFV and RVFV samples indicated that as the Cq values in the PCR decrease, more specific reads are obtained in the sequencing run (Table 1).

Genome assembly (de novo and map-to-reference) was successfully performed for samples with low Cq values (Table 2), and identities of more than 98% of the investigated viruses with reference sequences (Appendix A) were achieved in all target genomes (segments). Full genome sequences could be generated for all samples that showed Cq values of less than 22 in the respective qRT-PCR. Samples with Cq values of 23 to 27 showed varying results depending on the virus sequenced and the origin of the sample (e.g., generation of the whole genome of a WSLV cell culture sample with Cq 25). For the two samples with Cq values of 30, only a few specific reads were found (Table 1 and Table 2). In comparison to samples of animal origin, more specific reads were produced when sequencing cell culture isolates (Table 1). For NSDV, good coverage was achieved only for the cell culture sample (Cq = 18), whereas the bovine sample (Cq = 23) did not yield any sequencing results (Table 2). Identity levels (KMA) to the reference sequences (Table 2) ranged from 77.45% to 99.9%, while the coverage varied much more, from 1.19% to 100% (Table 2). Similar to the mean reading quality (Q), these two values were lower for samples with higher Cq values and/or for samples of animal origin.

## 4. Discussion

In recent years, third-generation sequencing with nanopores using MinION devices has become a reliable alternative and/or complement to second-generation sequencing techniques. Due to its small size and lower acquisitional costs, the device can be a valuable game changer, improving diagnostic capacities both in the field and in well-equipped laboratory facilities. Presequencing enrichment is a crucial aspect of sample preparation. In this context, virus-specific whole-genome PCRs are considered the gold standard, since due to their high specificity, whole-genome sequences can be obtained even with lower viral loads in the samples. However, more or less complex primer mixes have to be prepared depending on the virus to be sequenced. In some cases, very genetically diverse viruses such as CCHFV share only 70–80% of genetic identity among various strains [34], thus requiring different primer mixes for each strain amplification. Furthermore, the large amounts of viral amplicons produced by whole-genome PCRs can bear a potential risk of laboratory contamination. Another approach for a broad enrichment of viral genomic RNAs in cell culture and animal samples is the so-called SISPA technique, which allows a more open-view approach due to its nonselective amplification. In the studies herein described, we have therefore assessed the suitability of this nonspecific enrichment method as a preamplification step for MinION sequencing of different arboviruses occurring in Africa and compared the MinION sequencing results obtained from different sample types. 

A comparison of different samples of animal origin of CCHFV and RVFV showed that the highest number of specific reads was found in samples with lower Cq values, whereas the number of reads declined with increasing Cq values (Table 1). In contrast, no specific reads were obtained for an NSDV sample with Cq = 23, while more than 300,000 reads were found for WSLV with a Cq value of 25. Due to the limited comparability of different PCR assays, the Cq value can only be used as a vague indicator of the expected sequencing data outcome. However, the results of this study indicate that good sequencing performance can be expected for samples with Cq values below 22 when utilizing SISPA as a preamplification step for MINION sequencing. Besides a quantitative benchmark, sample quality should also be considered. Time and storage conditions of RNA samples strongly affect the quality of the nucleic acids [35]. In the case of field samples, a considerable amount of time can elapse from collection to the transportation/actual analysis in the lab, often making it difficult to fully maintain the cold chain. 

The quality of the reads and results obtained (Table 2) generally correlated with the number of reads generated for each sample (Table 1), i.e., the lower the Cq value, the better the quality of the results. Moreover, samples derived from cell culture supernatants performed very well for most of the viruses studied, especially in the case of WSLV and MIDV (Table 1). That might be explained by the higher degree of purity of cell culture supernatants (less foreign and interfering DNA, less nucleases, etc.) and the fact that the samples can be further processed immediately without longer transport distances. All cell culture samples also resulted in a better mean read quality compared to the respective animal samples (Table 2).

In general, the primer set of Peserico et al. [15] appeared to be more efficient, which was to be expected since the SISPA protocol used was designed for this primer pair and was not specifically adapted for the other primer set [29]. Interestingly, RVFV seems to be the only one of the six viruses examined for which the primers of Chrzastek et al. [29] performed better. Based on our data, it is rather difficult to determine whether the length of the 5′ tag N, the difference in the barcode sequence, or the fine-tuning of the original protocol using the primer set of Peserico et al. was responsible for those findings. The possibility of including reads that were unclassified in the first iteration resulted in a higher number of specific reads that ranged from 13% to 100% (Table 1). Since every specific read is valuable for assembling the target DNA/RNA alignment using nonspecific primers for sample enrichment, this can be a very helpful supplement. The dual primer set approach and consideration of unclassified reads also resulted in considerably better coverage and depth for all viruses. Therefore, to increase data yield in multiplexed sequencing runs, it seems advisable to sequence samples in duplicates (preferably with two different primer sets) if sufficient RNA material is available and to use the unclassified reads. It has to be mentioned that unclassified reads can solely be used if only one sample of the same origin (alone or in duplicate) is applied in one sequencing run (e.g., two samples of CCHFV from the same tick or two samples of CCHFV from the same cell culture supernatant). Unclassified reads cannot be distinguished when multiple samples of the same virus but of different origin have been sequenced in the same run (e.g., CCHFV from a tick and CCHFV from cell culture supernatant). 

In this study, it has been shown that with good sample quality, the use of SISPA amplification and MinION sequencing can provide a nearly complete genome sequence of the virus in most cases. Regardless of read quality and coverage, very good identity levels (between 90–100%) were achieved for all viruses when comparing them with reference sequences in the public database. Even for the RVFV cell culture sample, excellent identity levels of 98.4–99.9% were obtained despite a comparatively low coverage of 11.29–24.97% (Table 2B).

In summary, this study demonstrates and underlines the broad applicability of enrichment with SISPA for MinION sequencing. As the method allows the generation of viral (full) genomes without the need for virus-specific whole-genome PCRs, the main application of SISPA consists in the sequencing of a broad range of pathogens previously detected by different PCR assays to obtain an initial overview of the genetic diversity inside the sample panel. This makes it particularly interesting for emerging or neglected viruses that do not have a large history of published whole-genome primer protocols (e.g., MIDV or WSLV) and also for laboratories that are less well equipped. Nevertheless, enrichment by virus-specific whole-genome PCR would result in a better sequencing result for samples with poorer quality or a higher Cq value. Additionally, SISPA could be used in more open sequencing approaches (metagenomics) to identify yet unknown infectious agents. However, the initial quality of the samples represents the main limiting factor for a successful sequencing run. If an initial screening of the sample in a virus-specific qRT-PCR is possible, the Cq value can be taken as a rough parameter. 

## Figures and Tables

**Figure 1 pathogens-11-01502-f001:**
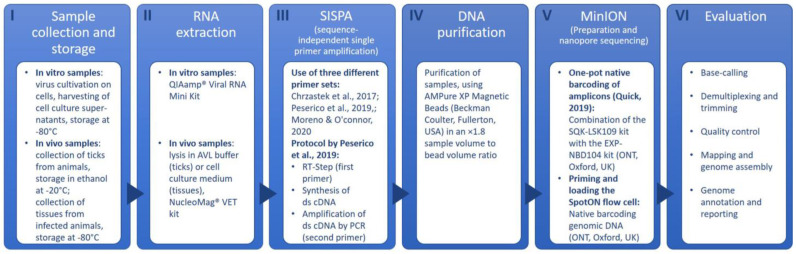
MinION sequencing workflow [15,29,30].

**Table 1 pathogens-11-01502-t001:** Total and virus-specific read counts of all examined samples by using two different primer sets [15,29] and by combining results of both primer sets and unclassified reads. The increase (%) in specific reads while using unclassified reads is indicated in parentheses.

Virus	Sample Type	Cq Value	Primer	Total Reads	Specific Reads/Segment		
					S	M	L
**CCHFV**	Animal origin	19	P	2,492,000	223	305	2533
			C	2,528,000	20	344	2314
			P+C+U	6,320,000	275 (+13%)	746 (+14%)	5725 (+18%)
		27	P	1,373,327	14	-	25
			C	1,816,806	4	-	27
			P+C+U	4,498,185	23 (+28%)	-	73 (+40%)
		30	P	4,196,000	1	-	2
			C	3,656,000	-	-	6
			P+C+U	9,336,000	1	-	9 (+13%)
	Cell culture	21	P	128,140	46	55	209
			C	187,246	1431	1078	4049
			P+C+U	1,757,226	2209 (+50%)	1737 (+53%)	6691 (+57%)
**RVFV**	Animal origin	20	P	716,000	22	125	168
			C	1,456,000	520	518	2012
			P+C+U	3,472,000	639 (+18%)	766 (+19%)	2567 (+18%)
		24	P	152,000	-	3	5
			C	633,319	11	4	23
			P+C+U	2,089,371	15 (+36%)	14 (+100%)	33 (+18%)
		30	P	432,000	-	-	-
			C	80,000	-	-	-
			P+C+U	1,996,000	-	-	-
	Cell culture	20	P	105,499	14	62	69
			C	126,457	14	83	48
			P+C+U	1,673,796	47 (+68%)	233 (+61%)	221 (+81%)
**NSDV**	Animal origin	23	P	28,000	-	-	-
			C	32,000	-	-	-
			P+C+U	1,496,000	-	-	-
	Cell culture	15	P	288,704	426	71	3830
			C	48,531	1	-	19
			P+C+U	1,644,776	511 (+20%)	88 (+24%)	4696 (+22%)
**DUGV**	Animal origin	20	P	4,044,000	25	66	215
			C	2,960,000	26	98	240
			P+C+U	8,444,000	55 (+8%)	180 (+10%)	523 (+15%)
	Cell culture	15	P	1,555,504	3199	13,744	44,478
			C	43,232	43	171	516
			P+C+U	2,906,788	4073 (+26%)	17,811 (+28%)	57,423 (+28%)
**WSLV**	Cell culture	25	P	2,493,349	219,001		
			C	82,025	4496		
			P+C+U	3,607,384	296,123 (+33%)		
**MIDV**	Cell culture	19	P	459,245	303,809		
			C	65,279	3884		
			P+C+U	1,556,534	400,767 (+30%)		

P = Peserico, Marcacci, Malatesta, Di Domenico, Pratelli, Mangone, D’Alterio, Pizzurro, Cirone, Zaccaria, Cammà and Lorusso [15]. C = Chrzastek, Lee, Smith, Sharma, Suarez, Pantin-Jackwood and Kapczynski [29]. U = unclassified reads. - = no data available. **Total reads** = the total number of reads at the end of the sequencing run that is included in the downstream analysis. **Specific reads/segment** = the number of reads that align to a known reference genome/genome segment. Consensus sequences were visualized in Geneious Prime 2021.0.1 (Biomatters Ltd., Auckland, New Zealand).

**Table 2 pathogens-11-01502-t002:** Overview of the most important quality parameters of the SISPA-based MinION sequencing results. Reference sequences used for identity levels can be found in Appendix A.

**(A) CCHFV**												
			**Coverage (%) and Depth**			**Mean Read Quality (Q)**	**Read Length N50 (bp%)**			**Identity Levels in Percent (KMA)**		
**Sample Type**	**Cq Value**	**Primer**	**Gene Segment**				**Gene Segment**			**Gene Segment**		
			**S**	**M**	**L**		**S**	**M**	**L**	**S**	**M**	**L**
Animal origin	19	P	91.9/4.13	91.0/7.37	44.93/1.76	11	1.5	8.1	11.1	99.8	99.5	99.6
		C	42.03/1.1	99.08/7.13	32.95/2.75	12.4	1.6	7.5	10.0	99.9	99.7	99.4
		P+C+U	97.4/4.86	99.71/14.39	92.04/10.26	11.5	1.8	9.4	12.1	99.9	99.8	99.6
	27	P	9.08/0.18	-	8.12/0.15	7.2	1.1	-	9.1	99.4	-	99.3
		C	8.42/0.20	-	8.46/0.17	7.0	1.06	-	4.2	99.6	-	99.5
		P+C+U	10.46/0.21	-	9.55/0.28	7.1	1.18	-	2.6	99.7	-	99.9
	30	P	3.42/0.05	-	4.83/0.05	-	0.7	-	2.1	98.1	-	99.5
		C	-	-	6.45/0.06	-	-	-	3.1	n	-	99.1
		P+C+U	2.84/0.05	-	7.42/0.07	7.7	0.6	-	3.6	99.1	-	99.7
Cell culture	21	P	9.78/1.18	7.6/2.24	18.12/2.15	12.90	2.5	7.1	14.1	98.1	98.7	99.1
		C	62.03/1.24	79.08/7.83	72.95/4.75	14.66	4.6	5.6	11.2	99.1	99.4	99.6
		P+C+U	94.4/7.86	91.71/24.39	96.04/14.26	17.02	8.8	9.9	19.2	99.9	99.7	99.7
**(B) RVFV**												
			**Coverage (%) and Depth**			**Mean Read Quality (Q)**	**Read Length N50 (bp%)**			**Identity Levels in Percent (KMA)**		
**Sample Type**	**Cq Value**	**Primer**	**Gene Segment**				**Gene Segment**			**Gene Segment**		
			**S**	**M**	**L**		**S**	**M**	**L**	**S**	**M**	**L**
Animal origin	19	P	1.29/0.11	58.34/0.58	35.22/0.35	10.86	1.2	9.2	10.1	98.7	99.1	99.2
		C	76.23/1.3	67.05/1.14	93.52/5.18	7.42	1.5	8.6	11.3	99.4	99.3	99.3
		P+C+U	74.97/4.42	99.90/5.17	99.90/130.2	7.51	1.9	8.5	14.1	99.8	99.7	99.9
	27	P	-	41.48/0.1	37.71/0.11	7.10	-	8.2	11.3	-	98.9	99.1
		C	2.86/7.4	43.46/0.2	42.85/0.63	8.96	1.5	7.4	14.3	99.3	99.4	99.4
		P+C+U	3.55/8.2	58.06/0.8	49.77/0.78	7.89	1.9	5.6	16.4	99.7	99.8	99.8
	30	P	-	-	-	-	-	-	-	-	-	-
		C	-	-	-	-	-	-	-	-	-	-
		P+C+U	3–7.36	-	-	7.01	1.1	-	-	98.1	-	-
Cell culture	21	P	11.29/0.21	14.34/0.48	5.22/0.75	13.4	2.1	7.2	11.1	98.4	99.1	99.4
		C	16.23/0.3	27.05/1.24	23.52/4.18	15.84	1.8	9.6	17.5	98.2	99.5	99.8
		P+C+U	24.97/2.42	32.90/3.27	28.90/5.2	13.86	1.7	7.7	16.6	99.5	99.9	99.9
**(C) NSDV**												
			**Coverage (%) and Depth**			**Mean Read Quality (Q)**	**Read Length N50 (bp%)**			**Identity Levels in Percent (KMA)**		
**Sample Type**	**Cq Value**	**Primer**	**Gene Segment**				**Gene Segment**			**Gene Segment**		
			**S**	**M**	**L**		**S**	**M**	**L**	**S**	**M**	**L**
Animal origin	23	P	-	-	-	-	-	-	-	-	-	-
		C	-	-	-	-	-	-	-	-	-	-
		P+C+U	-	-	-	-	-	-	-	-	-	-
Cell culture	18	P	90.9/7.36	85.96/9.54	89.81/7.34	12.4	2.4	7.5	9.9	99.1	89.4	99.3
		C	4.1/0.02	-	9.4/0.56	7.1	-	-	13.4	-	-	87.1
		P+C+U	99.9/37.15	95.96/99.51	99.81/87.36	11.5	3.7	4.6	12.9	99.62	92.5	99.54
**(D) DUGV**												
			**Coverage (%) and Depth**			**Mean Read Quality (Q)**	**Read Length N50 (bp%)**			**Identity Levels in Percent (KMA)**		
**Sample Type**	**Cq Value**	**Primer**	**Gene Segment**				**Gene Segment**			**Gene Segment**		
			**S**	**M**	**L**		**S**	**M**	**L**	**S**	**M**	**L**
Animal origin	20	P	1.29/0.20	4.34/0.45	1.22/0.79	8.9	1.8	8.2	11.1	98.9	99.4	99.4
		C	1.23/0.30	7.05/1.23	2.52/0.18	7.2	2.4	6.5	17.2	99.0	99.6	99.6
		P+C+U	4.97/1.42	8.90/1.25	3.07/0.13	8.4	7.5	9.1	14.5	99.4	99.8	99.9
Cell culture	15	P	91/101.71	89.21/200.02	92/211.36	15.6	3.3	8.2	13.6	99.2	98.9	99.1
		C	8.97/2.42	9.90/1.36	8.07/2.13	7.45	2.6	7.4	17.9	98.1	99.1	99.4
		P+C+U	100/142.74	99.98/222.99	100/234.36	14.1	4.7	4.6	14.5	99.36	99.63	99.88
**(E) WSLV**												
**Sample Type**	**Cq Value**	**Primer**	**Coverage (%) and Depth**			**Mean Read Quality (Q)**	**Read Length N50 (bp%)**			**Identity Levels in Percent (KMA)**		
Cell culture	25	P	96.4/1125.2			18.2	20.1			94.45		
		C	71.4/74.5			17.1	9.9			85.06		
		P+C+U	100/1388.25			18.6	18.6			99.59		
**(F) MIDV**												
**Sample Type**	**Cq Value**	**Primer**	**Coverage (%) and Depth**			**Mean Read Quality (Q)**	**Read Length N50 (bp%)**			**Identity Levels in Percent (KMA)**		
Cell culture	19	P	91.86/1556			17.8	21.4			89.32		
		C	59.42/42.36			18.9	11.4			77.45		
		P+C+U	94.47/1975.37			16.5	25.6			91.75		

P = Peserico, Marcacci, Malatesta, Di Domenico, Pratelli, Mangone, D’Alterio, Pizzurro, Cirone, Zaccaria, Cammà and Lorusso [15]. C = Chrzastek, Lee, Smith, Sharma, Suarez, Pantin-Jackwood and Kapczynski [29]. U = unclassified reads. - = no data available. **Total reads** = the total number of reads at the end of the sequencing run that is included in the downstream analysis. **Specific reads/segment** = the number of reads that align to a known reference genome/genome segment. **Depth** = the ratio between the total number of bases yielded by sequencing and the size of the genome. **Genome coverage** = the average number of reads that align to a known reference genome/genome segment. **Mean read quality** = the probability of a base being called incorrectly. A higher score indicates that a sequence is actually correct, and a lower score indicates that the sequence is more likely to be incorrect. **Read length N50** = the length of the shortest read in the group of longest sequences that together make up (at least) 50% of the nucleotides in the sequence set (based on the median and mean length of a set of sequences). **Identity levels in percent** = the number of nucleotide matches in the alignment (aligned with known reference genome, matched or mismatched).

## Data Availability

All data generated or analyzed during this study are included in this published article.

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
