# Peer review of "Whole-Genome Sequencing of Six Neglected Arboviruses Circulating in Africa Using Sequence-Independent Single Primer Amplification (SISPA) and MinION Nanopore Technologies"

_pathogens, 2022, doi:10.3390/pathogens11121502_

Round 1

Reviewer 1 Report

I have only minor comments/corrections.  

Section 2.1.5. MIDV and WSBV.  It looks like WSBV is incorrect and should be replaced with WSLV.  This should be corrected throughout the text and in the tables and figures. the same goes for WSVB (line 146).

Lines 316 - 325.  Can you provide a brief explanation of why the use of unclassified runs in combined data analyses from two different primer sets results in a higher number of specific reads?  Also, the sentence at the end of the paragraph (lines 323, 324) starting from "and to use only one sample of the same virus per sequencing ...  "  It seems to me that it repeats the statement in the same sentence (“to sequence samples in duplicates”), only in a confusing way.  I suggest deleting or re-phrasing it.   

Author Response

Dear reviewer,

we appreciate the positive feedback and comments. We have implemented your comments and hope that you are satisfied with the result.

Section 2.1.5. MIDV and WSBV.  It looks like WSBV is incorrect and should be replaced with WSLV.  This should be corrected throughout the text and in the tables and figures. the same goes for WSVB (line 146).

 A: Thanks a lot for the advice. We have corrected the corresponding errors accordingly.

Lines 316 - 325.  Can you provide a brief explanation of why the use of unclassified runs in combined data analyses from two different primer sets results in a higher number of specific reads?  Also, the sentence at the end of the paragraph (lines 323, 324) starting from "and to use only one sample of the same virus per sequencing ...  " It seems to me that it repeats the statement in the same sentence (“to sequence samples in duplicates”), only in a confusing way.  I suggest deleting or re-phrasing it.   

A: We rephrased the sentence and have included an additional paragraph in the M&M (LL. 243-249) and the discussion section (LL. 447-453) that describes the use of "unclassified reads" in more detail. We hope that we have cleared up the ambiguities.

Reviewer 2 Report

Comments on Schultz et al., “Whole-genome sequencing of six neglected arboviruses circulating in Africa using Sequence-Independent Single-Primer-Amplification (SISPA) and MinION nanopore technologies”

This is an interesting manuscript that describes the application of SISPA coupled with MinION long-read sequencing to identification of various viruses that are clinically and/or agriculturally important in laboratory samples and a limited number of clinical/field isolates.  Although there are very good reasons to attempt this approach, for which the amplification step does not rely on viral-specific primers, the significant weakness of this paper is that it is presented in an almost-entirely technical manner, without much context for the more general audience for the journal Pathogens.  In its current form, the work might be suitable to a more specialized journal. At a minimum, the manuscript will require more description of the rationale for the research, a more general and complete description of the protocols used (with more discussion of the contrasts with other approaches and the knowledge gaps or technical obstacles that this approach will fill), and a frank discussion of the viability of this approach for field samples based on the results obtained.

Specific points:

It should be made more apparent in the abstract that the work represents proof-of-concept that these viral genomes could be identified in laboratory samples by SISPA, rather than a broad screen for these viruses among field isolates.

Although many reading the manuscript will know this, the term “Cq” should be defined where it is first used.

Because the SISPA method represents the foundation of sample preparation to obtain the long reads by MinION, it should be described rather than just referenced.  Some additional discussion should also be included to bring out why specifically this approach offers advantages for this application over other sequencing approaches.

Describing and discussing the differences between the two SISPA primer sets and amplification strategies is especially important because of the divergent results emanating from the sequencing of the different libraries.

The Materials and Methods section 2.3 on analysis of the sequencing data is not particularly useful to many readers of Pathogens, as the analysis tools are only named and referenced.  For an audience that is not entirely bioinformaticians, it would be useful to at least briefly describe the steps of the analysis.

A better description of the parameters reported in the tables is required in the Results, table legends, or both.

Because of the apparently variable ability for this approach to detect viral genomes in some samples that are shown to contain viral genomes by RTqPCR, it is also of interest to evaluate the possibility of unexpected positive detections of viral genomes.  For example, it is not clear whether viral genomes could be detected in control samples (e.g., were RVFV genomes detected in any NSDV-positive samples, or from uninfected cells?)

Some more effort should be invested in both the Results and Discussion sections about what they say about the utility, feasibility, and implications of using this approach at a larger scale.  Some of the results are qualitatively intuitive, e.g., that samples pre-screened with RTqPCR and having lower Cq values would yield more/ better quality results at the MinION stage.  However, the failure to identify viral genomes in samples with lower thresholds of virus (by RTqPCR) or from more complex mixtures (animal samples) or lower quality samples should be more thoroughly addressed.  If the bottom line of this research is whether or not this approach should be used to screen for various viral infections, then a realistic/ pragmatic assessment of these results is in order.  The paper ends with a rather equivocal conclusion on this.

Author Response

Dear reviewer,

Many thanks for your positive feedback and critical comments. We have tried to take up your concerns and implemented them in the manuscript, in order to improve the quality of the manuscript significantly. We understand that the subject of the paper is very specific and may not address the typical readership of MDPI pathogens. Therefore, we have tried to present the topic in a more general way. Nevertheless, we would like to fairly emphasize that we submitted the manuscript as part of the Special issue " Molecular Diagnostics of Emerging Pathogens" and therefore hope that the paper is suitable for publication in the journal despite the higher degree of specialization.

Specific points:

It should be made more apparent in the abstract that the work represents proof-of-concept that these viral genomes could be identified in laboratory samples by SISPA, rather than a broad screen for these viruses among field isolates.

 A: We appreciate your comment and understand your concern. We have therefore reworded the abstract accordingly to prevent any misunderstandings. (LL. 16; 20-25).

Although many reading the manuscript will know this, the term “Cq” should be defined where it is first used.

 A: Thanks for the remark. We have added the definition of “Cq” (namely “quantification cycle”) in the text (L. 112).

Because the SISPA method represents the foundation of sample preparation to obtain the long reads by MinION, it should be described rather than just referenced.  Some additional discussion should also be included to bring out why specifically this approach offers advantages for this application over other sequencing approaches.

A: In response to your feedback, we have added a detailed description of the SISPA method both in the introduction (LL. 72-81) and the M&M section (LL. 192-212). We also attempted to better highlight the benefits of SISPA in the discussion (LL. 462-466).

Describing and discussing the differences between the two SISPA primer sets and amplification strategies is especially important because of the divergent results emanating from the sequencing of the different libraries.

A: Thank you for the comment. We have highlighted the differences between the two primer sets again in detail and pointed out that the same amplification protocol was used for both primer sets (LL. 193; 209-211).

The Materials and Methods section 2.3 on analysis of the sequencing data is not particularly useful to many readers of Pathogens, as the analysis tools are only named and referenced.  For an audience that is not entirely bioinformaticians, it would be useful to at least briefly describe the steps of the analysis.

A: We have tried to better explain some steps to make them more accessible to the general readership of the Journal (LL. 253-271). We hope that the section is now more comprehensible.

A better description of the parameters reported in the tables is required in the Results, table legends, or both.

A: We have added a comprehensive description of each parameter below the tables.

Because of the apparently variable ability for this approach to detect viral genomes in some samples that are shown to contain viral genomes by RTqPCR, it is also of interest to evaluate the possibility of unexpected positive detections of viral genomes.  For example, it is not clear whether viral genomes could be detected in control samples (e.g., were RVFV genomes detected in any NSDV-positive samples, or from uninfected cells?)

A: Thank you for this legitimate question. Given the way the MinION technology works, it is always possible that individual reads from other samples will show up in very small amounts in other samples. The main reasons for this are incorrect barcoding and incorrect base calling/transcription of the electrical potentials into nucleic acid sequences. However, this phenomenon is "generally" known and can be neglected due to very small quantities. We did not find any remarkably high amounts of unexpected reads in our samples. We hope that we were able to answer your question in a satisfying way.

Some more effort should be invested in both the Results and Discussion sections about what they say about the utility, feasibility, and implications of using this approach at a larger scale.  Some of the results are qualitatively intuitive, e.g., that samples pre-screened with RTqPCR and having lower Cq values would yield more/ better quality results at the MinION stage.  However, the failure to identify viral genomes in samples with lower thresholds of virus (by RTqPCR) or from more complex mixtures (animal samples) or lower quality samples should be more thoroughly addressed.  If the bottom line of this research is whether or not this approach should be used to screen for various viral infections, then a realistic/ pragmatic assessment of these results is in order.  The paper ends with a rather equivocal conclusion on this.

A: Thanks for the helpful comment. We have rewritten the last part of the discussion (also in the context of the other comments) to better highlight the advantages and disadvantages of SISPA in combination with MinION. In this way, we hope that the bottom line of the paper will become more straightforward.

Round 2

Reviewer 2 Report

I feel the authors did a very thoughtful job of responding to prior suggestions and criticisms.